# Mucoadhesive Marine Polysaccharides

**DOI:** 10.3390/md20080522

**Published:** 2022-08-15

**Authors:** Irina M. Yermak, Viktoriya N. Davydova, Aleksandra V. Volod’ko

**Affiliations:** G.B. Elyakov Pacific Institute of Bioorganic Chemistry, Far Eastern Branch, Russian Academy of Sciences, Prospect 100 Let Vladivostoku 159, 690022 Vladivostok, Russia

**Keywords:** mucoadhesive, chitosan, carrageenan, alginate, mucoadhesive interactions, methods of mucoadhesive

## Abstract

Mucoadhesive polymers are of growing interest in the field of drug delivery due to their ability to interact with the body’s mucosa and increase the effectiveness of the drug. Excellent mucoadhesive performance is typically observed for polymers possessing charged groups or non-ionic functional groups capable of forming hydrogen bonds and electrostatic interactions with mucosal surfaces. Among mucoadhesive polymers, marine carbohydrate biopolymers have been attracting attention due to their biocompatibility and biodegradability, sample functional groups, strong water absorption and favorable physiochemical properties. Despite the large number of works devoted to mucoadhesive polymers, there are very few systematic studies on the influence of structural features of marine polysaccharides on mucoadhesive interactions. The purpose of this review is to characterize the mucoadhesive properties of marine carbohydrates with a focus on chitosan, carrageenan, alginate and their use in designing drug delivery systems. A wide variety of methods which have been used to characterize mucoadhesive properties of marine polysaccharides are presented in this review. Mucoadhesive drug delivery systems based on such polysaccharides are characterized by simplicity and ease of use in the form of tablets, gels and films through oral, buccal, transbuccal and local routes of administration.

## 1. Introduction

Mucoadhesion is defined as the ability of materials to adhere to the soft mucosal surface that lines the gastrointestinal, tracheobronchial, reproductive and ocular systems [1]. It is presented as close a interaction at the interface between the drug and the mucosa. Mucoadhesion occurs between different types of mucous membranes and dosage forms (solid, viscous and liquid). Mucous membranes are moistened surfaces lining the walls of the digestive tract, the inner part of the eyelids, nasal and oral cavities and genitals [2,3,4,5]. Mucous tissue is a gate to the body for foreign substances and ensures the absorption of substances necessary for the body as well protection from harmful substances.

The potential for mucoadhesion in drug delivery was assessed and recognized back in the early 1980s, when Nagai and colleagues demonstrated the applicability of viscous gel ointments and mucoadhesive tablets for drug administration in the oral cavity [6,7], as well as indirect improvement in the bioavailability of the nasally administered peptide in the polymer medium [8]. Advantages associated with the use of mucoadhesives in drug delivery include an increasing dosage form residence times, improved bioavailability, reduction and simplification of the frequency of drugs administered for the effectiveness of therapy, as well as the ability to target certain body areas and tissues [5,9]. In addition, drugs boarded through the mucous membrane without oral direction often avoid the metabolism associated with its passage through the gastrointestinal tract, and also benefit from better penetration through the mucous membrane compared to the relatively low permeability of the transdermal route [5,10].

The moistened surface and the constant movement of the mucous tissue hamper the strong fixation and long-term retention of traditional drugs [11]. Therefore, overcoming the protective and mucosa barrier function is the basis for solving bioavailability problems. The adhesion of mucoadhesive dosage forms on the mucous membranes allows for reducing its total administered dose, both with systemic administration and local application [12,13,14].

The mucous membrane is covered with a viscoelastic gel layer, the mucus layer, which mainly consists of water (95%). The most important functional components of the mucus layer are the mucin glycoproteins (mucins), which are responsible for viscoelastic properties and interactions with its environment. They are responsible for the gel-like structure and anti-adhesive properties of the mucous membranes [15,16].

Mucins are a group of large proteoglycans which together make up the mucus layer on luminal surfaces of epithelial organs. The mucin molecules consist of a linear polypeptide backbone and radially arranged oligosaccharide chains, and the glycosylated domains are enriched with serine or threonine, which are linked to oligosaccharides via bonding with N-acetylglucosamine and N-acetylgalactosamine. The un-glycosylated areas, which normally exist with both N- and C-terminal ends, contain a large number of cysteine-rich domains and charged amino acids. One of the most interesting features of mucin is the high level of glycosylation, which makes the mass fraction of carbohydrate higher than that of peptide [15]. A more detailed description of mucins, their structure and properties is presented in the review [17]. Dried mucin could be dispersed in aqueous medium because it contains numerous hydrogen bonding groups, e.g., the hydroxyl groups in the branched sugar chains, the amide groups in the backbone chains and some carboxylic or sulfate groups in the terminal segments of branch chains [18].

Studies of the interaction of natural polymers with mucins led to the establishment of the nature of the processes of mucoadhesion and made it possible to predict the mucoadhesive properties of many dosage forms. Several theories have been proposed to explain the mechanism of interaction of polymers and mucosal glycoproteins [5,16,19]. At the beginning, close contact between the adhesive polymer and biological tissue should occur, due to sufficient moistening of the superficial mucosa and swelling of the adhesive [16,18,20]. After this, the polymer chains diffuse into the gaps and loops of the glycoprotein network [14], and after mechanical interweaving, weak primary, mainly non-covalent bonds are established [5,9]. Since mucins contain a large number of hydroxyl groups, it is considered reasonable to assume that hydrogen bonds play a key role in the development of mucoadhesion. The nature of the bioadhesive polymer and the environment in which it is placed are important for the manifestation of the properties of mucoadhesion. A mucoadhesive polymer should be characterized by certain physicochemical properties, including hydrophilicity and mobility of macromolecules, sufficient for diffusion through both mucus and epithelial tissue [5,16].

Mucoadhesive polymers are typically hydrophilic nets that contain numerous polar functional groups. Therefore, such polymers interact with mucus not only through diffusion, but also through secondary chemical bonds, leading to the formation of weakly crosslinked networks. Some mucoadhesive polymers can not only increase the residence time of the dosage form at the injection site, but can also increase the permeability of the drug through the epithelium by changing the tight junctions between cells [5]. In order to exhibit good mucoadhesive properties, the polymer must have specific structural characteristics such as strong hydrogen bonding groups, strong anionic or cationic charges, high molecular weight (MW), chain flexibility and surface energy properties that promote spreading to mucus [2,21]. Excellent mucoadhesive performance is typically observed for polymers possessing charged groups or nonionic functional groups capable of forming hydrogen bonds with mucous [5,16]. The formation of hydrogen bonds among the functional groups of the polymers and mucosal layer also plays an important role. In general, the stronger the hydrogen bonding, the stronger is the adhesion. The success and degree of mucoadhesion bonding are influenced by various polymer-based properties, such as the degree of crosslinking, the chain length and the presence of various functional groups [9,22]. The functional groups responsible for such interactions include hydroxyl, carboxyl and amino groups. The mucoadhesiveness of weakly anionic carboxyl-containing polymers, such as poly(acrylic acid), poly-(methacrylic acid), carboxymethylcellulose, sodium alginate and poly[(maleic acid)-co-(vinyl methyl ether)], has often been related to the ability of carboxylic groups to form hydrogen bonds with oligosaccharide chains of mucins [5]. It was shown that for mucoadhesion to occur, polymers must have functional groups that are able to form hydrogen bonds above the critical concentration, and the polymer chains should be flexible enough to form as many hydrogen bonds as possible [23]. With the increase in the chain length of the polymers, there is an increase in the mucoadhesive property of the polymer. Flexible polymer chains help in the better penetration and entanglement of the polymer chains with that of mucosal layer, thereby improving the bioadhesive property. The flexibility of the polymer chains is generally affected by the crosslinking reactions and the hydration of the polymer network. The higher the crosslinking density, the lower is the flexibility of the polymer chains [24]. The mechanism by which a mucoadhesive bond formed depends on nature of the mucous membrane and mucoadhesive material, the type of formulation, the attachment process and the subsequent environment of the bond [16].

Mucoadhesive polymers may provide an important tool to improve the bioavailability of the active agent by improving the residence time at the delivery site. The various sites where mucoadhesive polymers have played an important role include the buccal cavity, nasal cavity, rectal lumen, vaginal lumen and gastrointestinal tract (Figure 1). In recent years, among mucoadhesive polymers, carbohydrate hydrophilic biopolymers have been attracting attention as transport systems for delivering drugs. Polysaccharides are widely studied natural macromolecules because they are ideal for various drug delivery applications that rely on their mucoadhesive properties. They are generally stable, nontoxic, hydrophilic and biodegradable and display reactive functional groups (e.g., carboxyl or amino groups) that promote mucoadhesion. These include the sodium salt of carboxymethyl cellulose (NaCMC) [25], chitosan [22] and pectin [26]. The polysaccharides present in different marine algae species have been gathering great interest, which is a reflection of the continuous growth of knowledge on chemical and biological activities of these compounds. These polymers have been increasingly studied over the years in this context, given their potential usefulness in applications that mainly involve the design of drug delivery systems. Potential applications of sulfated seaweed polysaccharides in drug delivery systems are discussed in the review by Cunha and Grenhas [27]. Despite the large number of works devoted to mucoadhesive polymers, there are very few systematic studies on the influence of structural features of marine polysaccharides on mucoadhesive interactions. The purpose of this review is to characterize the mucoadhesive properties of marine carbohydrates with a focus on chitosan, carrageenan, alginate and their use in designing drug delivery systems.

## 2. Methods of Mucoadhesive and Form of Mucoadhesive Systems

Mucoadhesion is often studied in in vitro experiments by investigating the possible interactions between the samples and mucin on a molecular or colloidal scale to pre-screen the formulation prior to ex vivo tests. The presence of molecular interactions can be characterized by various spectroscopic methods (nuclear magnetic resonance spectroscopy, infrared spectroscopy, X-ray photoelectron spectroscopy) [28,29]. The binding enthalpy of the interactions can be measured directly by isothermal titration calorimetry (ITC) [30,31], or the macromolecular interactions with mucin can be characterized semi-quantitatively by measuring the fluorescence depolarization of labeled polymers [32] and atomic force microscopy [33].

The interaction of polymers and mucin particles could result in the formation of aggregates of different sizes, which can be characterized by various methods. Turbidimetric titration and dynamic light scattering (DLS) are obvious methods to study mucin/polymer interactions [21,30], as the formation of aggregates causes a marked increase in turbidity and light scattering. In general, changes in particle size or mass due to interactions are found in studies of mucoadhesive properties based on, inter alia, viscometry [34], velocity ultracentrifugation [35], quartz crystal microbalance, small-angle neutron scattering (SANS) [28] and small-angle light scattering (SALS) [36].

The oral mucoadhesive systems could be prepared in various forms such as buccal adhesive patches [37], films [38] and tablets [39]. Conventional drugs applied to the oral mucosa are generally in the form of in situ gels [40], pastes [41] or mouthwashes [42]. These forms have some drawbacks due to causing a high level of drug concentration and being in the oral cavity for a short period of time [43].

Mucoadhesion and bioadhesion of hydrogels is the result of a combination of surface and diffusional phenomena that contribute to the formation of adequately strong interchain bridges between the polymer and the biological medium [44]. Polysaccharide gels are widely used in drug delivery due to their large surface area and payload. Carrageenan-based hydrogels are brittle in nature with high swelling ratios under physiological conditions [45,46,47]. Thermoreversible gelation, biocompatibility, tunable viscoelastic properties and simple gelation mechanism make carrageenan an ideal polymer for controlled drug delivery applications [48,49].

Polymer mucoadhesive films have been developed for use in dentistry and ophthalmology for the past decade [37]. The thin films are a novel drug delivery tool and have been identified as an alternative approach to conventional dosage forms. The thin films are considered to be convenient to swallow, self-administrable and a controlled dissolving dosage form, all of which make them a versatile platform for drug delivery. This delivery system has been used for both systemic and local action via several routes such as oral, buccal, sublingual, ocular and transdermal routes [38]. The mucoadhesive films have been used as a delivery platform for the transmucosal buccal delivery of Biopharmaceutics Classification System (BCS) Class II drugs particularly targeting the opioid analgesics such as fentanyl citrate, which is available with a trademark name such as Onsolis^®^/Breakyl^®^ for treating immense pain [50].

Mucoadhesive polysaccharides may prolong the residence of ophthalmic drugs in the precorneal area. Mucomimetic polysaccharides, such as xyloglucan (tamarind seed polysaccharide, TSP), hyaluronic acid (HA) and hydroxyethylcellulose (HEC), which are currently used in commercial artificial tears for the treatment of dry eye syndrome, may as well prolong the residence of ophthalmic drugs in the precorneal area by virtue of their mucoadhesive properties. The ability of a polymer to improve ocular bioavailability of drugs by adhering to the ocular surface and binding the drug to it is a more promising property than the polymer’s gelling ability power, so far as fluid solutions are better tolerated than viscous ones [51,52].

Mucoadhesive micro- and nano-carriers can be used on almost any mucous membrane and are currently drawing significant interest from developers of dosage forms. Micro- and nano-carriers have new advantages in the presence of mucoadhesive properties, such as increased bioavailability due to the large surface area per unit volume and high affinity for mucous membranes [53,54,55]. The complexity of the interactions between different polymer-based mucoadhesive dosage forms and the viscoelastic surface of the mucous membranes has the attention of various scientific groups, as evidenced by the large flow of information on this topic and the search for new natural and synthetic mucoadhesive polymers.

## 3. Mucoadhesive Marine Polysaccharides

Many natural polymers have mucoadhesive properties, although the underlying mechanisms are not fully understood [4,56]. Menchicchi and co-authors tested a series of polysaccharides differing in primary structure, MW, charge density, conformation and the degree of coil contraction in the presence of salt, reflecting their intrinsic chain flexibility. They carried out a systematic analysis of the mucoadhesive properties of polysaccharides using defined experimental conditions (i.e., initial viscosity of the polysaccharide and mucin solutions), sensitive equipment, and a combination of data from different molecular levels, i.e., macroscopic data from microviscosimetry experiments and nanoscopic data from scattering experiments. This allowed them to characterize mucin–polysaccharide interactions in detail according to the MW, charge and degree of contraction of the polysaccharide chain. The authors showed that dextran sulfate interacts with mucin, as revealed by the changes in viscosity, but the MW of the polysaccharide plays a prominent role in this interaction. Their results confirmed experiments based on DLS and isothermal titration calorimetry, showing that the interaction between protein and alginate characterizes the dependence of MW. Although MW is thought to play an important role in such interactions, chain flexibility determines the ability of high-MW polysaccharides to induce the contraction of mucin [56].

### 3.1. Chitosan

Mucoadhesive properties have been widely studied for chitosan, both dry as well as in the form of a solution. In this regard, the mucoadhesive properties of chitosan do not raise doubts even among supporters of a strict classification, excluding viscous-flowing mixtures that are retained on the mucous membrane due to rheological properties. The ability of chitosan to interact with mucosal epithelia has been investigated in vitro and in vivo and the interactions between chitosan and mucin have been studied in different model systems [33,34,57,58].

Various research groups have studied detailed mechanistic evidence that occurs between chitosan and mucin at the molecular level. Chitosan is a semisynthetic linear polysaccharide consisting of D-glucosamine and N-acetylglucosamine residues linked by β-(1 → 4) bonds [59]. Chitosan is a random copolymer obtained from the alkaline deacetylation of chitin, the main component of crustacean shells.

Chitosan macromolecules contain several possible binding sites for mucin in their repeating unit: -NH_2_ or NH_3_^+^-deacetylated groups, hydrophilic OH groups and hydrophobic acetyl groups of non-hydrolyzed sites. At pH below 4.5, glucosamine residues are almost completely protonated, and at physiological pH values, the pKa of chitosan ranges from 6.1 to 7.0, depending on the type of polymer and measurement conditions [60]. The cationic nature, linear conformation, moderate flexibility of the chitosan molecule [57,61] and the low isoelectric point of mucin, which determines its negative charge at the most physiological pH values, are prerequisites for the high mucoadhesive activity of chitosan, which is based on electrostatic interaction. Since sialic acids have a pKa value of about 2.6, an almost uncharged mucin layer can be expected in the apical layers of the stomach [62]. The interaction of chitosan with mucin is an exothermic, enthalpy-driven process [21].

The interaction between chitosan and porcine stomach mucin in the presence of different additives confirmed that electrostatic interactions are complemented by hydrogen bonding and hydrophobic forces when chitosan and mucin are mixed in an aqueous environment [22]. Along with hydrogen bonds and ionic interactions, the mucoadhesion of the polymer to mucin can also occur through the physical adhesion [63], for example, due to the attraction of water by the polysaccharide from the layer of the mucous gel participates in the manifestation of mucoadhesive properties [64].

A wide range of various physicochemical methods are used to study the mucoadhesive properties of chitosan. These include rheological studies [34], viscometric and turbidimetric measurements [34,53], analytical ultracentrifugation [65], analysis of mucosal glycoproteins, DLS and monitoring of change in zeta potential of the original mucin particles after mixing with polymer solutions [66]. Relatively recently, the BIACORE method was applied for measuring the mucoadhesive interaction between chitosan and mucin This assay allows the determination of the affinity and binding kinetics of a ligand for its receptor, detecting the real-time binding association and dissociation rates [67]. The interaction between chitosan and porcine stomach mucin is also suggested by in vitro methods based on tensile and shear measurements [44,68].

When studying the mechanisms underlying this process, particular attention has been paid to the influence of the structural properties of chitosan, particularly the degree and nature of acetylation, as well as MW on this process. The various experimental methods that have been used to study such interactions reflect the structural complexity of chitosan molecules and their behavior in solution.

Currently, data on the effect of MW on the mucoadhesive properties of chitosan are rather contradictory. Interaction effects measured by viscometry, light scattering and titration calorimetry are better when mucin binds to high-MW chitosan [21]. An increase in the MW of chitosan leads to stronger adhesion due to deeper interpenetration of polymer and mucous chains, which is facilitated by the length of the macromolecule [69]. At the same time, the maximum interpenetration of chitosan and mucin molecules was found for more flexible chitosan samples with a low MW [70]. Thongborisute et al. have shown that chitosans with MW of 150 kDa and 22 kDa interacted with mucin with the same reaction rate [67].

An important role in this interaction belongs to the degree of N-acetylation (DA) of chitosan, which determines its behavior in solution and, as a consequence, the nature of the complex that it forms with mucin [21]. Chitosan with DA 50% interacts with pig stomach mucin at a higher polymer/mucin ratio than low acetylated chitosan, which is associated with a decrease in the number of free amino groups in chitosan with an increase in its DA [22]. Moreno et al. showed that chitosan microparticles with DA = 3% have a higher affinity for mucin compared to chitosan microparticles with DA = 18%. It is assumed that this is due to the large number of free amino groups in chitosan molecules with a lower DA [71]. However, a significant difference in mucoadhesive properties of the particles with different DA was not observed in tests on intestinal tissue. It has been suggested that this may be due to the participation in mucoadhesion of other compounds present in the intestinal mucus layer, as well as the participation of hydrogen bonds and hydrophobic interactions in addition to electrostatic interactions [72].

Menchicchi with co-authors [21] revisited the interaction between biomedical-grade chitosan polymers with different properties (MW and DA) and the non-gelled purified soluble fraction of rehydrated crude porcine gastric mucin in dilute aqueous solutions. They provided insight into the mechanisms that determine the mucoadhesive properties of chitosan, particularly the role of the DA and MW of the chitosan polymer. The authors used a combination of microviscosimetry, zeta potential analysis, isothermal titration calorimetry (ITC) and fluorescence quenching to confirm that the soluble fraction of porcine stomach mucin interacts with chitosan in water or 0.1 M NaCl via a heterotypic stoichiometric process significantly influenced by the DA of chitosan.

Based on the complex behavior of chitosan in dilute solutions, the authors hypothesized that the chain flexibility, hydrophilic/hydrophobic interactions, and the presence of charges in the chitosan chain, which together affect its conformation, determines the way of interaction with mucin and other macromolecules in solution. In turn, the relationship between the intrinsic viscosity of the high-MW chitosan samples and the DA suggests that chitosan adopts a more compact hydrodynamic volume as the chain becomes more acetylated, probably due to the presence of more hydrophilic structures, suggesting that chitosan behaves like a hydrophobic polyelectrolyte chain the “pearl necklace model”. Moreover, high-MW and high-DA chitosan can interact with the same magnitude in a larger range of composition as compared to low-DA chitosan, whose maximum interaction occurred only at a narrow range of composition (Figure 2). The obtained data allowed the authors to assume that chitosan–mucin interactions are mainly due to electrostatic binding, supported by other forces (for example, hydrogen bonds and hydrophobic association) [21].

The interaction between chitosan and mucins depends on not only internal factors (MW, surface charge, conformation), but also on external factors under which the interaction occurs. The ionic strength of the medium affects the interaction between polymers. At a high salt concentration, macromolecules behave like neutral particles due to the shielding of charged groups. This leads to a decrease in the contribution of the electrostatic component to their binding [73]. However, the interaction in solutions with high ionic strength still exists and varies depending on MW of chitosan [56]. It is assumed that in this case, hydrophobic interactions between the hydrophobic regions of mucins and acetyl groups of chitosan come to the fore.

Morariu et al. [74] consider concentration to be the key factor that determines the interaction between mucin and chitosan (Figure 3). There is a so-called critical polymer concentration (C*), above which chitosan chains overlap and form tangled networks. In the case of mucins, due to their physiological functions and structure, at high concentrations they are also known to form tangled and gel networks [74]. Adding excess of the cationic polymer in the solution led to the subsequent disaggregation of mucin particles (Figure 3). These facts greatly limit the possibility of interaction with mucin glycoproteins.

Ch’Ng et al. showed that the pH of the solution is one of the main factors affecting the mucoadhesive properties of both anionic and cationic polymers [75]. When studying the force of detachment of chitosan tablets with low DA from the mucous membrane of pig stomach, it was found that the force of detachment decreases when passing from an acidic (pH 1) to a neutral (pH 7) medium [54]. The value of the work of adhesion of chitosan disks during detachment from the surface of the mucin layer was higher at pH 5.2 compared to pH 6.8, and did not depend on the MW [31].

The determination of binding parameters at various pH levels has shown that the mucin/chitosan binding constants at pH 3 are significantly higher than those calculated at pH 7. However, at pH 3, the average number of binding sites (n) and the apparent binding constant (Ka) decrease with increasing temperature. On the other hand, the number of chitosan binding sites on mucin molecules and the apparent binding constant at pH 7 increase as one goes from 28 °C to 45 °C. These results suggest that complex formation between these two macromolecules at pH 7 was more favorable and stable at higher temperatures [76].

At pH 7, the cooperativity of binding of mucin with chitosan is higher than at pH 3. It is assumed that tryptophan residues in the hydrophobic site of mucin participate in the interaction. At the same time, it was found that electrostatic interactions between chitosan and mucin take place only at pH below 5. At pH 7, direct bonds are not formed between mucin and mucin; however, strong associations arise as a result of random and non-localized events, such as nonspecific dispersive (hydrophobic) associations, leading to polymer interlacements [76].

Based on electron microscopy data, a spherical model of complexes of pig gastric mucin and chitosan in solution was proposed [77]. It was similar to oil-in-water micelles with the addition of a surfactant. The oil-like phase in this model is represented by acetylated and therefore hydrophobic chitosan sequences, while mucin acts as a surfactant, thereby stabilizing the hydrophobic domains from further aggregation [77]. Rheological studies of the chitosan–mucin interactions have shown that there are two types of rheological behaviors, depending on the concentration of the polysaccharide and the mass ratio of the components [70]. The first is characterized by a drop in viscosity and occurs at a high ratio of chitosan to mucin. The other gives positive rheological synergy and is observed in the presence of excess mucin. It is believed that only this type of interaction correlates with an increase in mucoadhesive activity in vivo [70].

Although these studies are only a model of the interaction between chitosan and mucin in solution and are an oversimplification of the phenomenon of mucoadhesion that occurs in vivo, the authors believe that the high affinity of chitosan to interact with mucin can shrink the mucin gel network and thus create large pores throughout the gel network (i.e., between the bundles of chitosan–mucin), as suggested for alginate oligomers [78]. This may facilitate the diffusion of chitosan-based nano and microparticles and/or bioactive molecules through the mucus layer.

With the development of instrumental methods, a significant number of works have appeared devoted to the study of the interaction of chitosan with mucous tissues or model mucin layers. The ability of chitosan to increase the permeability of mucous membranes for the absorption of biologically active substances has been proven in a number of works [22].

In our study [79], we used the freshly frozen surface of the small intestine of a pig as a mucosal tissue and showed that film based on chitosan possessed a moderate degree of swelling on the mucosal surface. We supposed that this result is probably due to the degree of crystallization of its chitosan structure, evidenced by the X-ray and its ability to form strong intermolecular associations. The study of the chitosan adsorption on a preformed mucin layer showed that this cationic polyelectrolyte causes structural changes in the adsorbed mucin layer, leading to its compaction. This makes the mucin–chitosan layers resistant to degradation by surfactants, which indicates the protective properties of chitosan for mucous surfaces [80]. Pettersson and Dėdinaitė [81] also noted the thickening of the mucin layer upon adsorption of chitosan on it. The authors noted that during the formation of mucin–chitosan and mucin–chitosan–mucin layers, strictly layered structures are not formed. However, interpenetration of polymers due to the interaction between oppositely charged sialic acid residues from mucin and amino groups from chitosan located on opposite surfaces was observed [81].

The content of sialic acids in mucosal secretions differs depending on the type of mucous membrane [57]. The interaction of chitosan with the mucus varies significantly depending on the type of biological tissue [65]. Experiments with human mucin extracted from different parts of the stomach, namely, the cardia, the body and the antrum, showed a significant difference in the binding affinity of chitosan to mucin from different regions of the stomach. The maximum binding was observed for mucin from cardia and amounted to more than 30%, while binding with mucin from corpus was associated with less than 5%, and a minimal value of binding was detected for mucin from the antrum region [65]. It was shown that the penetration rate of chitosan into the mucosal layer of the ileum is higher than into the mucosal layer of the duodenum and jejunum. This is thought to be related to the thickness of the mucous gel layer in these areas of the intestine [82].

Chitosan in combination with mucin can be used to create model coatings to simulate the mucous membrane [83]. This approach produces a much thicker gel that mimics natural mucus gel better than the “monomolecular” mucus layer that results from conventional adsorption of mucin from solution onto a solid substrate. The creation of an artificial model of mucus is interesting from the point of view of drug delivery, since it allows studying the features of drug penetration through the mucosal layer. Artificial mucus matrices also have potential as implant coatings to reduce friction [84].

To improve the mucoadhesive properties of chitosan, the incorporation of chemical groups that enhance the interaction with the components of the mucin membranes is used. Since hydrophobic interactions play an important role in mucoadhesion [22], the introduction of acyl substituents into the chitosan molecule can improve its mucoadhesion. One of the most advanced approaches is the introduction of thiol fragments [85]. Thiolated chitosan derivatives exhibit more pronounced mucoadhesive properties in comparison to the unmodified polymer due to the ability to form covalent bonds between the thiol groups of the polymer and the glycoproteins of the mucous layer [86]. However, an excessive increase in the mucoadhesive properties of chitosan derivatives can limit their penetration through the mucin layer due to strong contacts with components with chitosan derivatives and mucin glycoproteins [67]. In addition, unmodified chitosan is also capable of temporarily disrupting tight junctions between cells, increasing the absorption of drugs by the paracellular route when administered orally and by inhalation [87].

The quaternization of the chitosan amino groups is often used to increase the solubility and intervals of chitosan application. However, Snyman et al. [88] showed that the presence of quaternary ammonium groups reduces mucoadhesive properties. This may be due to conformational changes in the trimethyl chitosan derivatives [88], as well as to the loss of their ability to form hydrogen bonds [22].

An extensive study of the mucoadhesive properties of chitosan showed that the polymer exhibits a pronounced mucoadhesive effect in a wide pH range. A higher binding affinity for mucin is shown by chitosans with a lower DA. The interaction is carried out due to electrostatic forces, hydrophobic interactions and hydrogen bonds, allowing the chitosan to exhibit mucoadhesive properties in a wide range of concentrations, in solutions with high ionic strength, as well as when interacting with mucous tissues or model mucin layers. The introduction of hydrophobic thiol substituents significantly increases the mucoadhesiveness of the chitosan, while the quaternization of its amino groups to increase the solubility of the polymer reduces its mucoadhesive properties.

The chitosan has proved to be a safe excipient in drug formulations over the past decades. It has attracted attention as an excellent mucoadhesive in its swollen state and a natural bioadhesive polymer that can adhere to hard and soft tissues. Good adhesion was found in epithelial tissues and in the mucus coat present on the surface of the tissues. Chitosan-based mucoadhesive systems are widely used for drug delivery [89,90].

For example, Luppi et al. [91] obtained and characterized mucoadhesive nasal liners based on chitosan–hyaluronan polyelectrolyte complexes. They showed that chitosan improved gel performance in terms of mucoadhesion strength, rheological properties and drug release. Similar effects of chitosan have been reflected in the works of many other researchers and described in detail in a number of reviews [54,92,93,94].

Chitosan is also often used to coat other carriers to make them mucoadhesive, such as liposomes [95], which facilitates their penetration through the intestinal mucosa [96].

The coating of carriers with mucoadhesive chitosan leads to the irreversible adsorption of a significant number of modified particles due to electrostatic interactions compared to unmodified ones [97]. According to our data [98], treatment of neutral and anionic types of liposomes with chitosan increased their ability to attach and hold onto the surface of the mucosal tissue. At the same time, the mucoadhesive effect of anionic liposomes coated with chitosan is almost 1.5 times higher than that of neutral ones.

Depending on the applied forms, the mucoadhesive properties of the chitosan carriers can vary considerably. A high ratio of surface area to volume of a particle in the case of nanosystems will enhance the adhesion interactions with mucin in comparison with larger structures [4].

Nanoparticles based on chitosan and its derivatives are able to protect the drug from acid denaturation, enzymatic degradation, and prolong the residence time of macromolecules in the small intestine. It is believed that mucin is spontaneously adsorbed on the surface of chitosan nanoparticles [99], causing a decrease in their ζ-potential [100]. According to Andreani’s data, the adsorption of mucin on chitosan NPs is higher at pH 6.8 than at pH 2.0 [101]. The adsorption of mucin on chitosan microspheres obtained by ionotropic gelation using genepin was higher at pH 3.6 than at pH 5.6, which is associated with a greater percentage of charged amino groups at a lower pH [102].

Analysis of the adsorption isotherms indicates that the adsorption of mucin on chitosan nanoparticles occurs at several sites with different affinities. In this case, the penetration of mucin into the inner pores of nanoparticles is also possible. Based on these data, it was assumed that in vivo nanoparticles will be retained by mucin at several points of adsorption on the mucous membrane, as well as penetrate deep into the tissue of the interstitial space, which will ensure long-term retention of chitosan nanoparticles on the mucous membrane [90].

In the case of nanoparticles, it is quite easy to control their physicochemical properties (viscosity, charge and particle size) in the synthesis process and, therefore, to regulate the force of interaction of chitosan with mucus. Chitosan nanoparticles, due to their mucoadhesive properties, can extend residence time in areas of the mucosa where absorption occurs (e.g., the small intestine), and thus increase drug absorption and lead to increased bioavailability [103]. The positive charge of chitosan promotes interaction with mucus and cell membranes, thereby preventing the effect of reverse diffusion and facilitating internalization of drugs [104,105]. In addition to chitosan, nanoparticles containing various drugs, due to their mucoadhesive properties, can improve transmucosal permeability, thereby enhancing the transport of nanoparticles along the paracellular pathway, and also cause structural reorganization of proteins associated with tight junctions [18].

### 3.2. Alginic Acid

Alginic acid is a biodegradable and biocompatible anionic polysaccharide composed of β-D-mannuronic acid and α-L-guluronic acid residues, linked by 1,4-glycosidic bonds, the sequence and proportions of which are determined by the algae source from which it is obtained [106].

Aqueous mixtures of purified mucin and alginate (ALG) were shown to form weak viscoelastic gels under appropriate conditions. The effect of ionic strength on gel rheology was indicative of the presence of electrostatic interactions within the gel matrix. Based on the rheology, it was proposed that mixed mucin ALG gels are maintained by both heteropolymeric mucin–ALG interactions and homopolymeric mucin–mucin interactions [107].

The associative interaction of porcine submaxillary mucins with sodium ALG and the dependence of this interaction on the mucin’s concentration was evaluated by comparing the rheological properties of mixtures against those of pure ALG and mucin in dilute, semi-dilute and concentrated solutions [108]. In concentrated solutions containing higher proportions of mucin, substantial binding interaction of mucins with ALG was observed. This was not observed in mixtures containing a high proportion of ALG, suggesting that mucins possess relatively low numbers of interacting sites. Introduction of 3 mM Ca^2+^ ions to mucin-ALG mixtures enhanced the binding strength of ALG to mucin. The interaction between water-soluble polysaccharides–ALG and the soluble fraction of partially purified porcine gastric mucin was assessed by the change in viscosity [56]. The nature of this interaction depended on the MW of the ALG. So, the low-MW polymer (ALG4) induced a slight increase in viscosity, in contrast to its high-MW counterpart (ALG400). Synchrotron SAXS data for mucin–ALG mixtures in solution confirmed that low-MW ALG4 interacts with mucin, presumably involving positively charged patches of the protein globules (i.e., von Willebrand-like domains) without affecting the overall expanded conformational state of mucin. Such interactions, therefore, do not influence the bulk properties of the solution such as viscosity and hydrodynamic size [56]. DLS-NIBS measurements in mixed ALG–mucin systems revealed that mucin at pH 4.5 exists as highly polydisperse colloidal species (PDI > 0.5). Size changes of mucin did not observe in the presence of ALG4; however, with the addition of ALG400, a change in the size of the mucin was observed [56]. The interaction of mucin with both ALGs was confirmed by fluorescence spectroscopy. Based on the spectra obtained, the authors suggested that ALG400 has a higher ability for the interaction with mucin and the formation of larger particles. The analysis of ALG–mucin systems by DLS and viscosimetry revealed that the hydrodynamic radii of species formed in the mixed solutions were smaller than those of the individual components [108]. A reduction in the hydrodynamic volume of the species in solution was accompanied by a proportional reduction in the size of Alg400–mucin, as observed by DLS-NIBS, which is consistent with results showing that the overall dimensions of mucin change in the presence of high-MW ALG400.

Analysis of microviscosimetry, scattering and spectroscopy data allowed Menchicchi et al. [56] to propose the following model for the interaction between ALG and the double-globular comb structure of mucin as a function of ALG MW and chain flexibility. Mucin was described in terms of double-globular protein regions connected by highly glycosylated linkers. The carbohydrate residues of mucin contain functional groups (e.g., carboxylic and sialic acids) that are negatively charged at pH > 2.5, thus maintaining the expanded conformation of mucin by repulsive interaction. Although the overall net charge of mucin is negative, positively charged patches in the non-glycosylated globular regions containing histidine, arginine and lysine residues can represent potential sites for interaction with negatively charged polysaccharides. Low-MW and stiff polyanions (Figure 4) are likely to interact preferentially with these globular regions without influencing the preferred conformation of mucin. In contrast, high-MW polyanions are more flexible and are likely to bridge distant sites, thus influencing the conformation of mucin and favoring a reduction in the overall hydrodynamic volume. This reduces the availability of interacting sites for additional polymer molecules and also occupies multiple sites simultaneously, saturating the available sites more quickly than low-MW polyanions. Although MW is thought to play an important role in such interactions, chain flexibility determines the ability of high-MW polysaccharides to induce a contraction of mucin (Figure 4).

The use of mucoadhesive agents with sodium ALG in the preparation of ALG-based particulate matrices for oral drug delivery have already proved to achieve better drug absorption as well as bioavailability of encapsulated drugs to facilitate mucoadhesion of the drug-releasing matrices with the intestinal region over a long period [109].

The mixed gels of mucin and ALG have been studied by both small deformation and large deformation rheology, and the effects of temperature and ionic strength on small deformation rheology have been investigated. According to the authors, the ability of ALG to interact with mucin and promote mucin–mucin interactions may also have relevance to understanding the clinical problems associated with pulmonary infection with the ALG-secreting bacterium *Pseudomonas aeruginosa* in cystic fibrosis patients [107].

Alginic acid was evaluated as a potential conveyor in ophthalmic solutions for prolonging the therapeutic effect of carteolol. In vitro studies indicated that carteolol is released slowly from alginic acid formulations, suggesting an ionic interaction. The adhesive behavior of alginic acid solution was better than that of another polymer, hydroxyethylcellulose (HEC). Intraocular pressure measurements of rabbit eyes treated with a 1% carteolol formulation with or without alginic acid showed that this polymer significantly extended the duration of the pressure-reducing effect of carteolol to 8 h. The results of this study indicated that the alginic acid vehicle is an excellent drug carrier, well tolerated and could be used for the development of a long-acting ophthalmic formulation of carteolol [110].

The ionotropically crosslinked gelled linseed polysaccharide (LP) with calcium ALG mucoadhesive beads containing diclofenac sodium for controlled release of diclofenac sodium over a prolonged time was obtained. The LP–calcium ALG beads loaded with diclofenac sodium displayed pH-responsive swelling and excellent biomucoadhesivity prospects with the intestinal mucosal tissue in both acidic and alkaline pH [111].

The fenugreek seed mucilage (FSM) ALG mucoadhesive beads as a mucoadhesive material containing metformin HCl was proved as a potential mucoadhesive excipient in the development of controlled-release mucoadhesive beads for oral use. The optimized FSM-ALG mucoadhesive beads containing metformin HCl showed significant hypoglycemic effects in alloxan-induced diabetic rats over a prolonged period after oral administration [109].

Sosnik overviewed the most relevant applications of ALG microparticles and nanoparticles for drug administration by the oral route and discussed the perspectives of this biomaterial in the future [112].

### 3.3. Carrageenans

Compared with commonly used pharmaceutical polymers, such as chitosan, pectin and ALG, the mucoadhesive properties of carrageenan (CRG) are not frequently reported. Little is known about the influence of the structural particular of CRGs on the properties. The mucoadhesive behavior of CRGs has been invoked to account for the ability of CRG-based matrix to transport macromolecular drugs across mucosal epithelia.

CRGs are a family of linear sulfated galactans of red seaweeds and are appreciated for their structural diversity associated with a large panel of physico-chemical properties and biological activities. CRSs have a primary backbone structure based on alternating 3-linked β-D-galactopyranose and 4-linked α-D-galactopyranose residues, and several types of these polysaccharides are identified on the basis of the structure of the disaccharide repeating units, by the sulphation pattern and by the presence of 3,6-anhydrogalactose (AGC unit) as a 4-linked residue [113]. The three most industrially exploited types, namely, κ-, ι- and λ-CRG (Figure 5), are distinguished by the presence of one, two and three ester sulfate groups per repeating disaccharide unit, respectively. Among the commercially available CRGs, κ- and ι-types form three-dimensional networks of double helix, while λ-CRGs do not form gels [114]. Native CRGs always represent complex hybrid structures or are generally a mixture of galactans composed of different carrabiose types. The hybrid nature of CRGs at the molecular level is responsible for changes in biological and physico-chemical properties of CRGs compared with those of their homopolymeric ideal types. At present, CRG has already been included in United States Pharmacopeia 35-National Formulary 30S1 (USP35-NF30S1), British Pharmacopoeia 2012 (BP2012) and European Pharmacopoeia 7.0 (EP7.0), implying that CRG may have a promising future as a pharmaceutical excipient. According to the JECFA, only degraded CRGs were associated with adverse effects and should not be used as food additives.

Although polysaccharide mucoadhesion has been widely studied, the basis of these properties in CRGS remains unclear.

In our study, the panel of different methods was applied to study the mucoadhesive properties of CRGs [115]. To explore the mucoadhesive properties of CRG, porcine stomach mucin, which is similar to the mucin in the epithelium of the gastrointestinal tracts of humans, was used. The interaction between CRGs of different properties (MW, absent/or present AGC) with crude porcine gastric mucin in dilute aqueous solution (at pH 6.5) was studied by DLS, by measuring their surface potential, by turbidity methods and by electron microscopy (SEM and TEM [115,116].

To study the effect of the structural features of CRGs on their mucoadhesive properties, the following samples of CRGs were used: λ- and κ-CRG from *C. armatus* [117], κ/β-CRG from *T. crinitus* [118] and ι/κ-CRG from *A. flabelliformis* [119]. The structures of these CRGs were established using chemical and physico-chemical methods of investigation, such as spectroscopy and mass spectrometry. There are three main dissimilarities in primary structure: presence or absence of 3,6-AGC, molar ratio of galactose to 3,6-AGC, the number and the positions of sulphate groups in some D-galactose residues and regular or irregular (hybrid) structure of the carbohydrate chain of the polysaccharides (Figure 5).

The changes in ζ-potential values may be used as an indication of mucoadhesive properties of polysaccharides. The ability of CRG to interact with mucin was determined by measuring their surface potential [115,116]. A monomodal ζ-potential distribution was measured for the CRG–mucin mixtures. The addition of CRGs to solutions of mucin resulted in the formation of a mixture that changed the charge of mucin. At pH 6.5, the mean ζ-potential of mucin particles was −13 mV, much lower than that of CRG (the surface charges of the different types of CRG varied from −70 mV for κ-CRG to −66 mV for λ-CRG), and it was in good agreement with data reported by Sogias et al. for mucin at pH 7.0 [22]. The charge values of the particles formed by the mucin with CRG, at different component ratios, may have been due to partial charge compensation. Mucin glycoproteins form highly entangled networks of macromolecules that associate with one another through noncovalent bonds [9,16]. The decrease in the ζ-potentials of the mixtures of all types of CRGs with mucin observed at specific ratios probably resulted from the adsorption of polysaccharide on the mucin surface and diffusion of the polymer chains into the gaps and loops of the mucin glycoprotein mesh. With the increasing mucin content, a decrease in the charge of the CRG–mucin mixtures were observed, which agrees with the above suggestion.

Turbidity of mucin in the presence of all types of CRG is slightly increased, which may relate to an aggregation leading to increase in its particle size caused by interaction with polysaccharides [115]. DLS measurements of mucin in the presence of CRG have revealed that all polysaccharide types cause aggregation of particles of mucin, leading to agglomerates. The particle size of the mucin increased by almost four times in the case of κ- and κ/β-CRGs (Figure 6a), but only slightly for λ- and κ/ι-CRGs. The large aggregates observed after mixing CRG and mucin in deionized water revealed the association between CRG and mucin, probably by the H-bonding. Scanning and transmission microscopy confirmed the occurrence of interactions between mucin granules and CRG chains. CRG incorporated into the structure of mucin transformed the supramolecular organization, according to the TEM and SEM data [115].

The charge values of the particles formed by mucin with CRG at different component ratios were much less than those of the CRG alone, and much larger than the value of the mucin. Increasing the mucin concentration in the mixture with CRG resulted in a shift in the ζ-potential of the mixture to a more negative value. It should be noted that for the mixture of κ- and κ/β-CRGs with mucin, the greatest charge change is observed, which is probably due not only to a lower degree of sulfation of these polysaccharides, but also to the features of their macromolecular structure. To evaluate the mucoadhesive properties of CRG, modified texture analyzer and fresh frozen inner surface of the small intestine of the pig as a model of mucous tissue were used. The work of adhesion value recorded for CRGs significantly exceeds the values obtained for the interpolyelectrolyte complexes (IPECs) based on Carbopol TM or low-viscosity polysaccharide chitosan [120].

The κ- and κ/β-CRGs had the greatest adhesion force, and λ-CRG had the smallest, which may be due to their greater MW. The observed differences in mucoadhesive properties of different types of CRGs are possibly due to differences in primary structure and MW of polysaccharides. The MW mass of κ-CRG was more than three times greater than that of λ-CRG and twice as great as that of κ/ι-CRG. CRG with 3,6–AGC units interacted more strongly with mucin. The weaker interaction of λ-CRG with mucin can be explained by both the low MW of the polymer and the flexibility of the carbohydrate chains. Based on the calculation of the configuration statistics of single chains of CRGs, it was shown that the flexibility of κ- and ι-CRG chains is more than λ-CRG [121]. Chain flexibility has been suggested to maximize the formation of heterotypic contact points between the polymer and the corresponding part of the mucin molecule, thus promoting interpenetration and entanglement [16].

Furthermore, because of the ^1^C_4_ conformation of their 3,6-AGC units, κ- and ι-CRGs are able to undergo coil to helix conformation, the subsequent association of helices leading to their gelation [122]. X-ray diffraction studies on oriented fiber samples have shown that ι- and κ-CRGs form ordered helices [123].

λ-CRG differs from the other investigated CRG types by possession of a high degree of sulphation and the absence of 3,6-AGC. In λ-CRG, the conformation of both of the galactose residues of a disaccharide unit corresponds to the C_1_ chair conformation, and ordered helices are not formed by this structure [124]. The ordered structure of CRGs is probably more complementary or preferable to the supramolecular homogeneous network of mucin, which it forms at high pH.

The observed difference may be due not only to the primary but also the supramolecular structure of these polysaccharides. It is known that during contact with a mucous membrane, the polymers swell and therefore expose a maximum number of adhesive sites, which enables inter-diffusion and interpenetration of polymer chains and mucin network [125]. As it was found, in the macromolecules of CRGs, cavities are formed. In the case of κ- and κ/β-CRG, these cavities are the most massive [126]. The presence of these cavities probably contributes to a greater swelling of CRG on the mucin surface and its binding. In mucoadhesive terms, swelling is favorable as it not only allows greater control of drug release, but additionally, the swelling process increases the surface area for polymer/mucus interpenetration possessing excellent mucoadhesive characteristics due to the formation of strong hydrogen bonding interactions with mucin.

It has been proposed that interaction between mucin and polymer is a result of secondary bonding, mainly hydrogen bonding and physical entanglement [127]. To estimate the role of bonding forces in the interaction between oppositely charged mucin and CRGs, different additives were used [115]. The role of hydrogen bonding in the interaction of CRG with mucin was demonstrated using intermolecular hydrogen bond breaking agents such as urea [128]. The addition of CRG to mucin in the presence of urea (Figure 6a) is accompanied by a decrease in turbidity and a change in ζ-potentials in the case of κ- and κ/β-CRGs, whereas in the case of λ-CRG, it did not change. This indicated the participation of hydrogen bonds in the interaction of κ/β-CRG with mucin. It is known that the expanded nature of the polymer network contributes significantly to the strength of mucoadhesion [129]. κ/β-CRG appears to have an open extended conformation and a more favorable balance between available hydrogen bond sites as compared to the more compact λ-CRG structure. At the same time, in solution with urea, where hydrogen bonding is fully prevented and hydrophobic effects are weakened, λ-CRG still interacted with mucin, presumably via electrostatic forces. It is well known that addition of inorganic salts, for example, NaCl, is able to disrupt the interaction between oppositely charged synthetic polyelectrolytes [130]. The presence of 0.15 M NaCl suppressed mucin–polysaccharide interactions. The ability of NaCl to attenuate the putative interaction between polyions λ-CRG and mucin indicated that ionic interactions play a key role in this case (Figure 6b). The interaction between CRG and porcine gastric mucin in the presence of various additives confirmed that hydrogen bonds and electrostatic interactions are complemented when CRG and mucin are mixed in an aqueous medium. CRGs that contained the AGC units had high MW and exhibit a high density of available hydrogen bonding groups able to interact more strongly with mucin glycoproteins [115].

Due to their mucoadhesive properties, CRGs are widely used for drug delivery in various forms [39]. Upon cooling, and in the presence of an appropriate cation (K^+^, Ca^2+^), CRGs, particularly κ-type, undergo coil to helix transition and helical aggregation to form thermotropic and ionotropic hydrogels. CRGs are extensively reported as the excipients to prepare gel beads due to their easy gelling, thermo reversibility of the gel network and appropriate viscoelastic properties [48,131].

Modified κ-CRG/pectin hydrogel patches were fabricated for the treatment of buccal fungal infections. To enhance mucoadhesive properties of the polymers, the authors modified κ-CRG and pectin with two different thiolated agents (L-cysteine and 3-mercaptopropionic acid). The force of adhesion of the obtained composites were estimated by the authors as the most important parameter of mucoadhesion. Although the thiolated group contributed to adhesion enhancement, the greatest adhesive force showed CRG/pectin hydrogel patches composites with a predominance of CRG [132].

It has been shown that hydrogel of CRG is safe at ophthalmic level. The main goal in ocular treatment is to achieve a high concentration of the drug without ocular surface damage [133]. With this aim, several polymeric ocular formulations have been studied to improve the viscosity and/or mucoadhesivity and to increase the contact time of the drug with the ocular surface, among which hydrogels have been the most effective. It has been shown that ion-sensitive hydrogels based on gellan and κ-CRG are interesting as an advanced ophthalmic drug delivery system. Different proportions of these biopolymers were analyzed using a mixture experimental design evaluating their transparency, mechanical properties and bioadhesion in the absence and presence of simulated tear fluid. Tears induced a rapid sol-to-gel phase transition in the mixture, forming a consistent hydrogel. The solution composed of 80% gellan gum and 20% κ-CRG showed the best mechanical and mucoadhesive properties [134].

The improvement of bioavailability can be achieved with the employment of mucoadhesive polymers that prolong the precorneal residence time by interacting with the ocular mucin layer. Ion-activated in situ gelling systems have great potential as ocular drug delivery systems due to the presence of mono and divalent cations such as Na^+^, K^+^, Mg^2+^ and Ca^2+^ in eye tears. The interaction with salts present in the tears importantly affects the properties of the hydrogel contains κ-CRG [135].

Bonferoni with co-authors used λ-CRG to obtain CRG gelatin mucoadhesive systems for ion exchange based ophthalmic delivery [136]. As a model, an alkaline drug timolol maleate was chosen, with which λ-CRG, as an anionic polymer, is able to interact. The authors evaluated mucoadhesive properties by the force of the detachment between the films obtained on the CRG and the mucin dispersion. According to the results, the films on the λ-CRG did not show mucoadhesive properties. Only the combination of CRG with gelatin provided mucoadhesive properties of films containing a drug model. Their results are consistent with our data, which showed the very weak mucoadhesive properties of λ-CRG [115].

In a number of works, Vigani et al. developed gelling formulations based on κ-CRG for the treatment of oral mucositis and esophagitis induced by cancer therapies [137,138]. κ-CRG was chosen for its ability to gel in the presence of saliva ions. The formulations also contained hydroxypropyl cellulose employed as a mucoadhesive agent, and CaCl_2_ that was proved to enhance, at a low concentration (0.04% *w*/*w*), the interaction between κ-CRG and saliva ions. Different component concentrations were investigated in order to obtain formulations able to interact with saliva ions, producing a gel capable of adhering to the damaged mucosa. The developed formulations were characterized by an easy administration due to their low viscosity at room temperature, a protective action towards the mucosa related to their marked elastic properties at 37 °C, and mucoadhesion properties [138]. Later, in situ gelling formulation loaded with Hibiscus sabdariffa (HS) extract intended for the treatment of oral mucositis and esophagitis was developed. The presence of HS produced a lowering of the formulation viscosity at room temperature and enhanced κ-CRG interaction with saliva ions [138].

In our study, CRG and CRG/chitosan gel beads were used as a mucoadhesive controlled release delivery system for echinochrome A (Ech), the active substance of the drug Histochrome^®^ [139]. The beads were evaluated by their properties, such as morphology, mucoadhesive properties, swelling behavior, drug release in the simulated gastric fluid and tear fluid. CRG beads showed good mucoadhesive properties, which were evaluated using mucous tissue of the small intestine of the pig as a model and texture analyzer. CRG/Ech beads exhibited similar or slightly greater mucoadhesive properties compared to the drug-free system (CRG beads), which may be due to the ability of drug molecules to interact with polysaccharides. The low drug release from polysaccharide matrices was observed for CRG beads that contained 3,6-AGC units and had high MW. CRG beads containing Ech were coated with chitosan by layering. Chitosan slowed down the release of Ech, which can be explained by the formation of less porous beads coated with chitosan, according to SEM analysis of the surface of the CRG/chitosan beads. The surface properties of beads play an important role in drug transport through the mucosa. The presence of positive charges in chitosan promotes its interaction with the negative charges of mucus macromolecules, thereby enhancing bioadhesion. CRG beads may be proposed as possible suitable candidates for ophthalmic and oral delivery of Ech [139].

Inclusion of drugs in liposomes offers the potential for localized and sustained delivery to mucosal surfaces. We used CRG as a soluble matrix to incorporate echinochrome A (Ech) into liposomes. The interactions of liposomes containing CRG/Ech with porcine stomach mucin were determined by the DLS and SEM methods. The changes in the ζ-potential and size of the mucin particles were observed as the result of the interaction of liposomes with mucin. To evaluate the mucoadhesive properties of liposomes and the penetration of Ech on the mucosa, was used a fresh frozen inner surface of the small intestine of a pig as a model of mucous tissue. Liposomes containing CRG/Ech exhibited very good mucoadhesive properties—50% of Ech remained on the mucosa [116].

The combination of the polymers ι-CRG and hydroxypropyl methylcellulose was developed for the controlled release of acyclovir in order to obtain mucoadhesive vaginal tablets [39]. To establish the tablets’ capacity to adhere to the vaginal mucosa, the work and force necessary for detachment were assessed using the texture analyzer. It was shown that all the formulations can attach to the vaginal mucosa. According to the authors, this confirms the mucoadhesion of the studied polymers due to their adhesion mechanisms mediated by hydrogen bonds and electrostatic interactions. It is interesting that the formulations with one polymer had higher force and mucoadhesion work values, and that these values decreased when the formulations contained a combination of polymers. The formulation interacted with the vaginal mucosa and prolonged the residence time of the pharmaceutical dosage. The high mucoadhesive capacity of obtained samples allowed the formulation to remain in the vaginal area long enough to ensure the complete release of acyclovir [39].

We studied films obtained from different types of CRGs, chitosan and a three-layer (containing polyelectrolyte complex, CRG–chitosan (PEC)). Mucoadhesive properties of the films of CRGs were assessed by the ability of the films to swell on the mucous tissue and their adhesion/erosion after contact with the mucosa [79]. All studied CRG films exhibited mucoadhesive properties. Single-layer CRG films had a higher degree of swelling than chitosan film, probably due to the degree of crystallization of chitosan structure, which is evidenced from the X-ray and its ability to form strong intermolecular associations. The swelling of CRG films’ value correlated with the number of sulphates in polysaccharides and decreased in the λ > κ/β-CRG series, which can also be explained by the difference in their macromolecular organization. The inclusion of the active substance Ech in the films increased their swelling capacity on the mucosal surface [79].

The application of CRG in drug delivery is rapidly evolving due to its distinctive gelling mechanism, ample functional groups, strong water absorption and favorable physiochemical properties. Potential applications of sulfated seaweed polysaccharides in drug delivery systems are discussed in some reviews [27,47,52]. Cunha and Grenha [27] discussed potential applications of sulfated polysaccharides in drug delivery systems, with a focus on carrageenan, ulvan and fucoidan. General information regarding structure, extraction processes and physicochemical properties is presented, along with a brief reference to reported biological activities. For each material, specific applications under the scope of drug delivery are described, addressing in a privileged manner the particulate carriers, as well as hydrogels and beads, matrix tablets, fibers and films. The application of sulfated polysaccharides in targeted drug delivery, focusing with particular interest on the capacity for macrophage targeting. In conclusion, the authors note that in order to achieve the maximum potential of using such materials, in-depth research is needed on their chemical structure and physical and chemical properties. The optimization of extraction procedures, providing more pure biomaterials, is also a desired achievement. In fact, the uncertainty about structures and the difficulties in extraction are definitely the strongest limitations regarding the proposal of applications, thus preventing the progression of these materials to more advanced therapeutic solutions.

The review [52] discusses the available polysaccharides such as cellulose, hyaluronic acid, alginic acid and chitosan, as well as polysaccharide derivatives, choices for overcoming the difficulties associated with ocular drug delivery, and it explores the reasons for the dependence between the physicochemical properties of polysaccharide-based drug carriers and their efficiency in different formulations and applications. These polysaccharides have been successfully used to augment drug delivery in the treatment of ocular pathologies. The authors show that the properties of polysaccharides can be extensively modified to optimize ocular drug formulations and to obtain biocompatible and biodegradable drugs with improved bioavailability and tailored pharmacological effects. The review summarizes the current advances being made in the development of ocular drug carriers based on natural and semi-synthetic polysaccharides, with emphasis on the properties of polysaccharides as drug carriers and their biological interactions in the eye.

The administration of CRG hydrogels through various routes for drug delivery applications has been critically reviewed by Ramanathan Yegappan et al. [47]. The authors provide an overview of developing various forms of CRG-based hydrogels. CRG hydrogels are generally formed through thermoreversible gelation, ionic crosslinking as well as modification of CRG backbone with photocrosslinking methacrylate moieties. These modifications are further explored to fabricate different forms of CRG hydrogels with interesting features. The authors outline the application of these hydrogels not only pertaining to sustained drug release but also their application in bone and cartilage tissue engineering, as well as in wound healing and antimicrobial formulations. Finally, the authors conclude that CRG is versatile, promising biomaterial for a variety of bioengineering applications.

## 4. Conclusions

In this review, we analyzed and described the mucoadhesive properties of marine polysaccharides with a focus on chitosan, CRG and ALG. Mucoadhesion is defined as the ability of materials to adhere to the soft mucosal surface that lines the gastrointestinal, tracheobronchial, reproductive, ocular and nasal systems. The nature of the mucoadhesive polymer and the environment in which it is placed are important for the manifestation of the properties of mucoadhesion. Excellent mucoadhesive performance is observed for polymers possessing charged groups or non-ionic functional groups capable of forming electrostatic interactions and complemented by hydrogen bonding or hydrophobic forces with mucosal surfaces. Polymers such as chitosan, ALG and CRGs interact with mucus through diffusion, but also through secondary chemical bonds, which leads to the formation of weakly crosslinked networks. Due to their high ability to adsorb water, these polysaccharides can improve drug dissolution, thus increasing the oral bioavailability of poorly water-soluble drugs. The ability of marine polysaccharides to interact with mucin and mucosal epithelia that has been investigated in different model systems is presented in this review. Due to physical and chemical properties such as the ability to form gels and increase the viscosity of solutions, marine polysaccharides—i.e., chitosan, ALG and CRGs—can be processed into various forms, such as hydrogels, microspheres and films, and are widely used in the biomedical field. Mucoadhesive drug delivery systems based on such polysaccharides can significantly increase the effectiveness of the drug used and are characterized by simplicity and ease of use through oral, buccal, transbuccal and local routes of administration.

## Figures and Tables

**Figure 1 marinedrugs-20-00522-f001:**
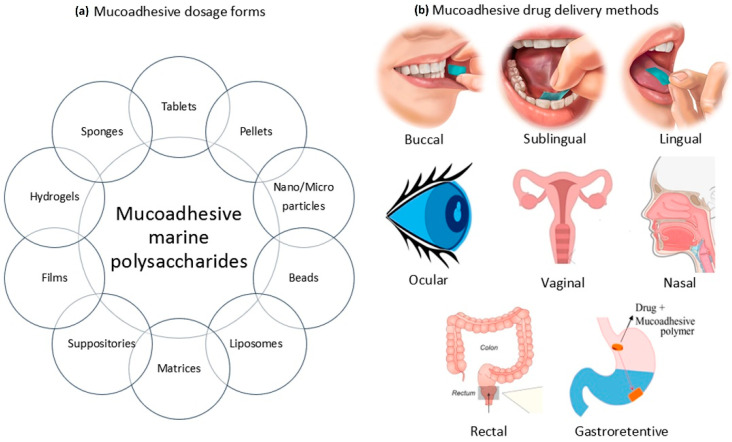
Mucoadhesive dosage forms based on marine polysaccharides and methods for their delivery: (**a**) different forms of mucoadhesive drugs; (**b**) delivery routes for mucoadhesive drugs.

**Figure 2 marinedrugs-20-00522-f002:**
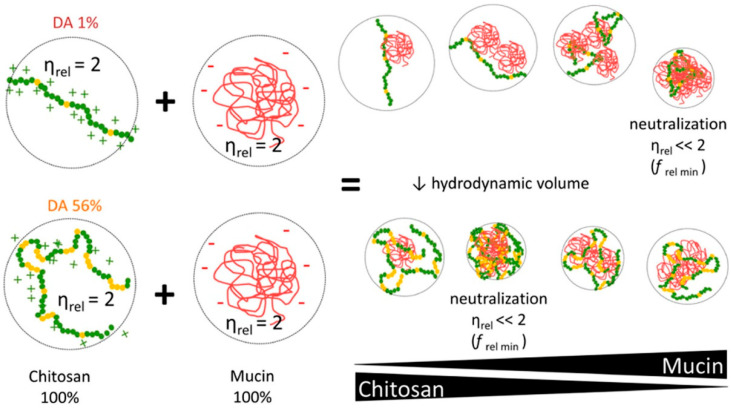
Influence of the chitosan DA on the interaction with mucin. Green and yellow dots indicate the chitosan molecule (green dots, glucosamine units; yellow dots, N-acetyl-D-glucosamine); the red line indicates the mucin molecule; η_rel_—relative viscosity; ƒ _rel min_—the point at which low-MW chitosan-containing systems show their maximum in viscosity reduction. Reprinted (adapted) with permission from [21], copyright © 2022 American Chemical Society.

**Figure 3 marinedrugs-20-00522-f003:**
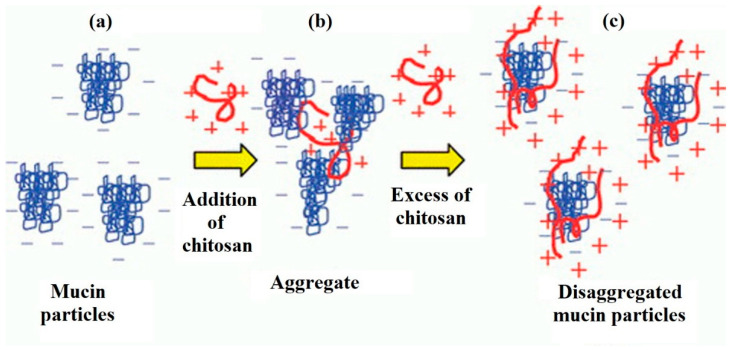
Influence of the chitosan concentration on the interaction with mucin. Aggregation/de-aggregation of mucin particles in the presence of a cationic polymer: (**a**) mucin dispersion in the absence of a polymer; (**b**) mucin dispersion in the presence of a small portion of a polymer; (**c**) mucin dispersion in the presence of excess polymer. Reprinted (adapted) with permission from [22], copyright © 2022 American Chemical Society.

**Figure 4 marinedrugs-20-00522-f004:**
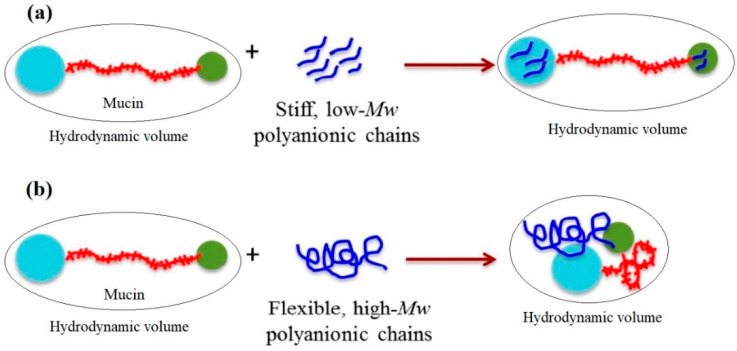
Model for the interaction between alginate and the double-globular comb structure of mucin as a function of alginate MW and chain flexibility. (**a**) Low-MW and stiff polyanions to interact with globular regions without influencing the preferred conformation of mucin, thus having a negligible impact on bulk properties such as size and viscosity; (**b**) high-Mw polyanions are more flexible and to bridge distant sites, thus influencing the conformation of mucin and favoring a reduction in the overall hydrodynamic volume. Reprinted (adapted) with permission from [56], copyright © 2022 American Chemical Society.

**Figure 5 marinedrugs-20-00522-f005:**
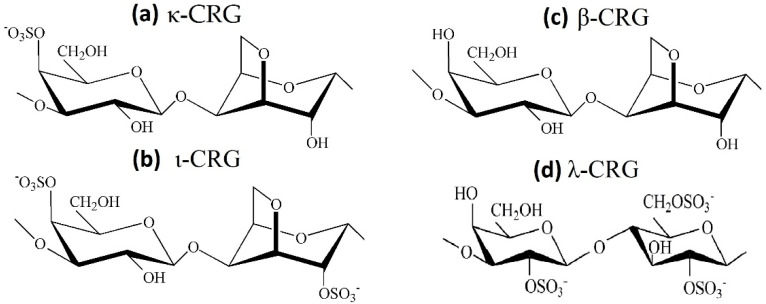
Chemical structures of the repeating disaccharide units: (**a**) κ-CRG; (**b**) ι-CRG; (**c**) β-CRG; (**d**) λ-CRG.

**Figure 6 marinedrugs-20-00522-f006:**
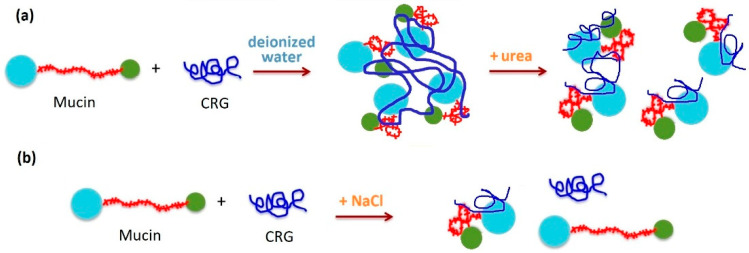
Model for the interaction between CRGs and mucin the as a function of role bonding forces in the interaction: (**a**) in deionized water, CRGs interact with mucin to form agglomerates. Urea as hydrogen bond breaking agents causes disaggregation of these aggregates. (**b**) The presence of 0.15 M NaCl suppresses mucin–polysaccharide interactions, although minor interactions of CRG and mucin were maintained under these conditions. The interaction between CRG and mucin in the presence of various additives confirms that hydrogen bonds and electrostatic interactions are involved. Scheme of mucin adapted with permission from [56], copyright © 2022 American Chemical Society.

## Data Availability

Not applicable.

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
