# Peer review of "Mucoadhesive Marine Polysaccharides"

_marinedrugs, 2022, doi:10.3390/md20080522_

Round 1

Reviewer 1 Report

Please address the comments included both in the original manuscript and the two-page document. 

Author Response

The authors thank the referee for a thorough and detailed analysis of their article. We feel that the comments allowed us to improve the quality of the manuscript. We have changed manuscript according to remarks.

General comments

  1. Revise the entire review for language correction

Corrections have been done

  1. Add more figures to display the different applications of these polysaccharides

Corrections have been done. Figure 1 has been added

  1. Write more details in the figure captions (figure 1, 2, and 3) to explain each figure clearly

Corrections have been done

  1. Page 1, line 23, the font size of the keywords needs to be adjusted

Corrections have been done

  1. Page 1-2, line 31 and 49, references should be added

Corrections have been done

  1. Page 2, line 68-70, this paragraph should be merged with the above paragraph

Corrections have been done

  1. Page 2, line 74 and 90, references should be added

Corrections have been done

  1. Page 3, line 96-103 Explain how the functional groups of the polymers could impact their mucoadhesive properties

Corrections have been done

  1. Page 3, line 105-108, this paragraph could be merge with the above paragraph

Corrections have been done

  1. Page 3, line 128-130, in vitro and ex vivo should be italic in the entire review not just in these two lines, please revise

Corrections have been done

  1. Page 4, line153, add a reference

Corrections have been done

  1. Page 153 what is CRG?

Corrections have been done

  1. Page 173, line 176, (gelformising) what is this word?

Corrections have been done. gelformising → gelling ability

  1. Page 4, line 183-185 could be merged with the above paragraph and line 188-189 could be merged with the paragraph below

Corrections have been done

  1. Page 4, line 191, “MW” is written without its name, please write the complete word before its abbreviation

Abbreviation previously deciphered: Line 94 - “cationic charges; high molecular weight (MW); chain flexibility; surface energy properties”

  1. Page 4, line 198, The mentioned study is very interesting, elaborate more about the findings
  2. Corrections have been done
  3. Page 4, line 199, the tittle should be on the same page with its paragraph

Corrections have been done

  1. Page 5, line 200, please rewrite the sentence

Corrections have been done

  1. Page 5, line 208, add more data and discussion about the mechanical properties of chitosan

The authors apologize for the erroneous spelling. The corresponding changes have been done.

  1. Page 5, line 214, what is the pKa of chitosan? and what is its pH/charge at the physiological pH?

Corrections have been done

  1. Page 5, line 232, what is the BIACORE method? Explain it in one or two sentences

Information has been added

  1. Page 5, line 245, please explain which Mw of chitosan showed a better binding to mucin

Corrections have been done

  1. Page 6, line 247, please define DA before writing it as an abbreviation

Corrections have been done.

  1. Page 6, line 255, wide range of what ?? it is not clear what do you want to say

The sentence has been removed

  1. Page 6, line 252-254, please revise it and explain it in more clear way to show which DA of chitosan resulted in a higher interaction with mucin

Corrections have been done.

  1. Page 7, line 302, what is this parameter? what the parameter decreases, please explain

Corrections have been done.

  1. Page 7 line 343-357 the three paragraphs could be merged

Corrections have been done.

  1. Page 9 line 403, It would be very useful if you add a paragraph that summaries all the factors affecting the interaction between the chitosan and mucin such as DA, MW, functional group, etc before starting to mention the drug delivery applications of chitosan

Information has been added

  1. Page 9, line 433, do you mean the synthesis process by “the process of obtaining”?

Corrections have been done.

  1. Page 13, line 583, Please add the zeta potential of CRGs before adding it to the mucin so the difference in charge become clear

Information has been added. “the surface charges of the different types of CRG varied from −70 mV for κ-CRG to −66 mV for λ-CRG”

  1. Page 15, line 691, More studies/examples could be added to highlight the applications of CRG in ophthalmic drug delivery applications

Corrections have been done

  1. Page 15, line 710, add more examples of CRG applications for the treatment of oral mucositis and esophagitis

Corrections have been done

  1. Page 16, line 765, the authors mentioned that “Potential applications of sulphated seaweed polysaccharides in drug delivery systems are discussed in some reviews [22,46,131].” Mention some examples of these applications in the review

Corrections have been done

  1. In the references, some DOI are missing, for example in reference number 6, 19, 34, 37, 74, 94, 97, 116, 125, 132, please revise

Information has been added.

  1. without DOI
  2. without DOI, full text → https://www.jocpr.com/articles/mucoadhesive-polymers-means-of-improving-the-mucoadhesive-properties-of-drug-delivery-system.pdf
  3. Pacheco-Quito, E.M.; Ruiz-Caro, R.; Rubio, J.; Tamayo, A.; Veiga, M.D. Carrageenan-based acyclovir mucoadhesive vaginal tablets for prevention of genital herpes. Mar. Drugs 2020, 18, 249, doi:10.3390/md18050249.
  4. without DOI, full text → https://www.ijpsonline.com/articles/mucoadhesive-buccal-drug-delivery--a-potential-alternative-to-conventional-therapy.pdf
  5. Volod’ko, A. V; Davydova, V.N.; Petrova, V.A.; Romanov, D.P.; Pimenova, E.A.; Yermak, I.M. Comparative Analysis of the Functional Properties of Films Based on Carrageenans, Chitosan, and Their Polyelectrolyte Complexes. 2021, 1–23, doi:10.3390/md19120704.
  6. without DOI, full text → https://lib.sevsu.ru/xmlui/bitstream/handle/123456789/9411/2019%20Т.4%2C%20â„–2.pdf?sequence=1&isAllowed=y
  7. Andreani, T.; Miziara, L.; Lorenzón, E.N.; de Souza, A.L.R.; Kiill, C.P.; Fangueiro, J.F.; Garcia, M.L.; Gremião, P.D.; Silva, A.M.; Souto, E.B. Effect of mucoadhesive polymers on the in vitro performance of insulin-loaded silica nanoparticles: Interactions with mucin and biomembrane models. Eur. J. Pharm. Biopharm. 2015, 93, 118–126, doi:10.1016/j.ejpb.2015.03.027.
  8. Mustafin, R.I.; Semina, I.I.; Garipova, V.R.; Bukhovets, A. V.; Sitenkov, A.Y.; Salakhova, A.R.; Gennari, C.G.M.; Cilurzo, F. Comparative Study of Polycomplexes Based on Carbopol® and Oppositely Charged Polyelectrolytes as a New Oral Drug Delivery System. Pharm. Chem. J. 2015, 49, 1–6, doi:10.1007/s11094-015-1211-2.
  9. without DOI, full text → http://europepmc.org/abstract/MED/3293807
  10. without DOI. It is book section.

Reviewer 2 Report

The purpose of this manuscript is characteristics of mucoadhesive properties the marine carbohydrates with a focus on chitosan, carrageenan, alginate and their use in designing drug delivery systems. It is a topic of interest to the researchers in the related areas but the paper needs very significant improvement before acceptance for publication. My detailed comments are as follows:

1. There are some small mistakes in this manuscript and need more corrections, One of the obvious errors is the line 47 “prevents”.

2. Are there any differences in drug delivery systems and adhesion properties of various forms of marine polysaccharides, such as microspheres, films, etc?

3. How do these marine polysaccharides achieve a safe and effective drug delivery system through their adhesion properties?

Author Response

The authors thank the referee for a thorough analysis of their article. We have changed manuscript according to remarks.

1.There are some small mistakes in this manuscript and need more corrections. One of the obvious errors is the line 47 “prevents”

prevents → hamper

  1. Are there any differences in drug delivery systems and adhesion properties of various forms of marine polysaccharides, such as microspheres, films, etc?

Various administration routes, such as ocular, nasal, buccal and gingival, gastrointestinal (oral), vaginal and rectal, make mucoadhesive drug delivery systems attractive and flexible in dosage form development.  Mucoadhesive drug delivery systems may be formulated as tablets, films, solid inserts, wafers, pessaries, suspensions, in situ gelling systems and sprays.

Mucoadhesive tablets are most commonly developed for oral use, also promising application in gynecology. The main limitation for their wide application is the size and shape, since there is a requirement, so that drug forms a close contact with mucosal surface. Unlike tablets, polymeric films are sufficiently plastic, to take the form of a subsurface, and also has a number of advantages over creams and ointments, since they can maintain an accurate dosage of drug in application area. The eye films containing polymer with improved mucoadhesive properties are a multi-layered example of a solid form for ophthalmic surgery. The mucoadhesive micro- and nanocarriers compared to other solids has more contact for an effective absorption of drug into the bloodstream and characterization greater bioavailability. Due mucoadhesive properties form dense contact with the surface of the mucosa, and easy penetration drug into areas with difficult access, capture and retention in the uneven structure of the mucous membrane. Hydrogels or three-dimensionally cross-linked networks of hydrophilic polymers are soft, elastic and porous materials capable of imbibing large quantities of water and resembling the properties of biological tissues. The gels have a limited propagation for drugs with a narrow interval therapeutic dose or for difficult areas, but it is convenient for ophthalmology, stomatology and gynecology. A gel-like ophthalmic delivery system, e.g. based on κ-carrageenan, is ion-activated in situ due to its ability to rapidly form gels when exposed to mono- and divalent cations (e.g., Na+, K+, Mg2+ and Ca2+, etc.) present in the lacrimal liquids.

  1. How do these marine polysaccharides achieve a safe and effective drug delivery system through their adhesion properties?

Mucoadhesion is defined as attractive interaction at the interface between a pharmaceutical dosage form and a mucosal membrane. The majority of these dosage forms incorporate polymeric excipients, which play a major role in their mucoadhesivity. Excellent mucoadhesive performance of marine polysaccharides observed for these polymers to due possessing charged groups or nonionic functional groups capable of forming hydrogen bonds with mucous. Marine polysaccharides are non-toxic, rich in source and easy to obtain. It has been proved to be biocompatible and biodegradable in human body. Based on the special characteristics of polysaccharides such as the strong charge and gelling properties, swelling capacity they have been use as mucoadhesive agents for controlled drug release and prolonged retention. The unique ability of polysaccharide hydrogels to swell in water and living tissue-like consistency make them significant candidates for developing various biomaterials and dosage forms. Application safe marine hydrophilic polymers lengthens the retention time of the drug form on the slick tissue, which leads to a gradual release of the active substance and better transferability from the side of the patient. The applications of hydrogels in biomedical and pharmaceutical sciences include soft contact lenses, drug delivery systems, and wound dressings. Mucoadhesive polysaccharides may prolong the residence of ophthalmic drugs in precorneal area. Polysaccharides, such as cellulose, hyaluronic acid, alginic acid, and chitosan, carrageenan as well as polysaccharide derivatives, have been successfully used to augment drug delivery in the treatment of ocular pathologies. The properties of polysaccharides can be extensively modified to optimize ocular drug formulations and to obtain biocompatible and biodegradable drugs with improved bioavailability and tailored pharmacological effects. The polysaccharides carriers and excipients are now being used to enhance ocular drug delivery, and several reviews have described the use of natural polymers for this purpose.

The mechanism by which a mucoadhesive bond  formed  depend on the nature of the mucous membrane and mucoadhesive material, the type of formulation, the attachment process and the subse quent environment of the bond.

The ion-activated matrix formulations (including dosages of KC and HPMC, and their combination) were designed and screened using fluidity and gelation capacity as indicators. The results indicated that the optimized matrix formulation caused no irritation to rabbit eyes. that ACV in situ ophthalmic gel matrix was prepared optimized through in vitro release test, stability studies and ocular irritation assay.

In this feature article we attempted to provide an overview of existing knowledge about mucosal membranes and mucins, mucoadhesion and mucoadhesive polymers, techniques used to characterize mucoadhesive properties of various dosage forms and also highlighted some recent developments in novel classes of mucoadhesive polymers.

Mucoadhesion and bioadhesion of biomaterials, and especially hydrogels, is the result of a combination of surface and diffusional phenomena that contribute to the formation of adequately strong interchain bridges between the polymer and the biological medium.

Round 2

Reviewer 2 Report

More summative graphs should be added in the section 2, section 3.3 and other right sections.

Author Response

Section 2, Lines 159-161 “Figure 1. Mucoadhesive dosage forms based on marine polysaccharides and methods for their delivery: (a) different forms of mucoadhesive drugs; (b) delivery routes for mucoadhesive drugs.”

Section 3.3, Lines 619-620 Figure 5. Chemical structures of the repeating disaccharide units: (a) κ-CRG; (b) ι-CRG; (c) β-CRG; (d) λ- CRG.”

Section 3.3, Lines 656-662 “Figure 6. Model for the interaction between CRGs and mucin the as a function of role bonding forces in the interaction: (a) in deionized water, CRGs interact with mucin to form agglomerates. Urea as hydrogen bond breaking agents causes disaggregation of these aggregates; (b) The presence of 0.15 M NaCl suppresses mucin–polysaccharide interactions, although minor interactions of CRG and mucin were maintained under these conditions. The interaction between CRG and mucin in the presence of various additives confirms that hydrogen bonds and electrostatic interactions is involved.”